# Exosomal microRNAs and Progression of Nonalcoholic Steatohepatitis (NASH)

**DOI:** 10.3390/ijms232113501

**Published:** 2022-11-04

**Authors:** Xiaoyan Qi, Jinping Lai

**Affiliations:** 1Department of Endocrinology and Metabolism, South China Hospital, Health Science Center, Shenzhen University, Shenzhen 518116, China; 2Department of Pathology, Immunology and Laboratory Medicine, University of Florida College of Medicine, Gainesville, FL 32601, USA; 3Department of Pathology and Laboratory Medicine, Kaiser Permanente Sacramento Medical Center, Sacramento, CA 95825, USA

**Keywords:** exosomes, extracellular vesicles (EVs), microRNAs (miRNAs), nonalcoholic steatohepatitis (NASH), mitochondria, apoptosis, diagnosis, targeted therapy

## Abstract

Nonalcoholic fatty liver disease (NAFLD)/metabolic associated fatty liver disease (MAFLD) is becoming a public health problem worldwide. Steatosis as the simple form and nonalcoholic steatohepatitis (NASH) as its progression form are commonly seen in liver biopsy specimens from patients with obesity, diabetes, hyperlipidemia, hypertension, and the use of certain drugs. Patients with NASH and advanced fibrosis were associated with increased risks of liver-related complications, including hepatocellular carcinoma (HCC). However, the mechanisms regarding the progression from simple steatosis to NASH fibrosis remain incompletely understood. Because NASH-caused liver injury is a complex process and multiple cell types are involved, intercellular communication is likely mediated by extracellular vesicles. Exosomes are a type of small extracellular vesicles and contain various cellular molecules, including proteins, messenger RNAs (mRNAs), and microRNAs (miRNAs). MiRNAs are short, non-coding RNA species that are important post-transcriptional regulators of gene expression and may play an important role in the pathogenesis of NALFD/NASH. In this article, we review the articles about NASH and exosomal miRNAs published in the most recent English literature through PubMed search and discuss the most recent criteria for histological diagnosis, pathogenesis from steatosis to NASH, roles of exosomal miRNAs in NASH pathogenesis and progression, as well as their potential in future clinical diagnosis and treatment for patients with NASH.

## 1. Introduction

Nonalcoholic fatty liver disease (NAFLD)/metabolic associated fatty liver disease (MAFLD) is becoming a public health problem worldwide [1]. Recent epidemiologic studies have shown that NAFLD has become the most common chronic liver disease in the United States, and it was estimated to have a prevalence of 34% in the general population based on screening using liver proton magnetic resonance spectroscopy [2,3,4,5,6]. Steatosis and steatohepatitis are the most common morphologic patterns of liver injury in patients with NAFLD, which are commonly seen in liver biopsy specimens from patients with obesity, diabetes, hyperlipidemia, hypertension, and the use of certain drugs. Nonalcoholic steatohepatitis (NASH) is recognized as the progressive form of NAFLD. The well-recognized histologic features of NASH include steatosis and ballooned hepatocytes, mixed lobular and portal inflammation, and zone 3 perisinusoidal fibrosis [7,8,9,10]. A recent clinical prospective study led by NASH Clinical Research Network (CRN) showed that NASH-caused advanced fibrosis stages (F3 and F4) were associated with increased risks of liver-related complications, including hepatocellular carcinoma and death [11].

So far, no current FDA-approved therapies have been used for the treatment of this disease [12]. Lifestyle modifications, including commonly used diet control, are always difficult to maintain. However, several drugs that are currently available for other indications have been studied in phase 2b trials for NAFLD, such as vitamin E and pioglitazone, and are endorsed by current guidelines as a possible treatment in selected patients with NASH [13,14]. Vitamin E is promising for the improvement of steatosis and histological benefit in patients without diabetes or cirrhosis [15]. Reportedly, combination therapy of vitamin E with pioglitazone can achieve histological improvement [4]. Majzoub et al. conclude that compared to placebo, Lanifibranor, Obeticholic acid, Pioglitazone, and Vitamin E were statistically significantly better in achieving ≥1 stage of fibrosis improvement [16]. Larger comparative randomized controlled trials (RCTs) are warranted to further establish the comparative efficacy of different interventions for NASH in demonstrating ≥1 stage improvement in fibrosis and/or NASH resolution. Finding appropriate therapeutic targets is now more urgent than ever before.

Because NASH-caused liver injury is a complex process in that multiple cell types are involved, intercellular communication is likely mediated through the interaction of extracellular vesicles. Exosomes are a type of small extracellular vesicles ranging from 30 nm to 120 nm in size. They contain various cellular molecules, including proteins, messenger RNAs (mRNAs), and microRNAs (miRNAs). A novel mechanism of cell-to-cell communication is proposed through recent studies on exosomes and microvesicles since they carry lipids, proteins, and various species of RNA that can trigger a myriad of responses in target cells [17]. MiRNAs are short, non-coding RNA species whose primary functions are to inhibit gene expression [18]. Mature miRNAs bind to complementary seed sequences in the 3′-UTR (untranslated region) of messenger RNAs (mRNAs), leading either to their degradation or inhibition of their translation by restraining their access to the translational machinery. Furthermore, some miRNAs may enhance gene expression, providing an additional layer of complexity in the miRNA molecular functions [19]. MiRNAs can interfere with all aspects of cell activity, such as proliferation, differentiation, metabolism, apoptosis, and carcinogenesis [20]. Because they are protected from degradation by RNases, in addition to being resistant to the extremes of temperature and pH, they can be a potential biomarker, both diagnostic and prognostic, for several diseases [21,22]. A recent study found that miRNAs regulate lipid metabolism, inflammation, and fibrosis in NAFLD through the post-transcriptional silencing of the target genes by base-pairing to partially complementary sites in the 3’-UTR of target mRNA [23]. In this article, we review the articles about NASH and exosome miRNA published in the most recent English literature through PubMed search and discuss the recent histological diagnosis and pathogenesis from steatosis to NASH, the role of exosome miRNAs in NASH pathogenesis (reported in vitro and in vivo studies and limited clinical studies) and their potentialities for future clinical diagnosis and treatment.

## 2. Histologic Diagnosis of NASH

Liver biopsy has been used as the golden standard for the diagnosis of NASH [2,3,4,5,6]. Although some advanced imaging techniques, such as MRI, could be helpful for the diagnosis of NASH, our recent clinical trial showed that changes in hepatic fat content measured by MRI do not predict treatment-induced histological improvement of steatohepatitis [3]. The histologic hallmarks of NASH are steatosis (Figure 1A), ballooned hepatocytes, and inflammation (Figure 1B) [2,7]. Steatosis is the simplest form of fatty liver disease characterized by the presence of fat droplets in hepatocytes without other abnormal findings. This finding is sometimes termed simple steatosis. Based on the NASH CRN criteria, steatosis is evaluated at low magnification, and a visual estimate is made of the proportion of parenchyma involved by macrovesicular steatosis (Figure 1A). Steatosis is graded as grade 0, 1, 2, and 3, with macrovesicular steatosis present in less than 5%, 5–33%, 34–66%, and more than 67% of the hepatic parenchyma, respectively [7,8,9,10]. The steatosis can be mainly in zone 3 and zone 1 or may involve all zones equally in a panacinar pattern. Ballooned hepatocytes and lobular inflammation (Figure 1B) are the other two key features to distinguish steatohepatitis from simple steatosis. A definitive diagnosis of NASH requires ballooned hepatocytes in addition to steatosis and lobular inflammation [2,7,8,9,10]. Ballooned hepatocytes are swollen hepatocytes (>1.5 times of the adjacent normal hepatocytes) with or without voluminous clear to rarified cytoplasm and small bits of eosinophilic Mallory–Denk bodies (Figure 1B). Other histologic features seen in patients with steatosis or steatohepatitis include hepatocyte apoptosis (acidophilic body) (Figure 1C), cytoplasmic glycogenosis, megamitochondria (Figure 1D), and iron deposition, usually of mild degree.

There are three histologic scoring systems developed for grading steatohepatitis, including steatosis, activity, and fibrosis (SAF); the Brunt staging system; and the NASH CRN NAFLD activity scoring system [24]. The NASH CRN system is modified from the Brunt staging system and is broadly used in pathology practice [2,7,8,9,10]. Based on the NASH CRN criteria, the overall grade of steatohepatitis in this scoring system is expressed as the sum of the steatosis grade, the lobular inflammation grade (0, 1, 2 and 3, with 0, <2, 2–4, and >4 foci of lobular inflammation per 10× field, respectively), and the ballooning grade (0, 1 and 2, with none, few and many ballooned hepatocytes, respectively). The total is referred to as the NAFLD activity score (NAS), and it ranges from 0 to 8, depending on the findings. Regarding the staging of fibrosis for steatohepatitis, in 1999, Brunt et al. proposed a fibrosis staging system for NAFLD to take into account the unique perisinusoidal/pericellular patterns of fibrosis in the central zone seen in NASH (Figure 2A) [9]. This system was revised in 2005 and incorporated into the NAS stage system with modifications. The modified stage 1 includes stage 1a with mild perisinusoidal/pericellular fibrosis that requires trichrome stain to recognize, stage 1b with moderate perisinusoidal/pericellular fibrosis that can be seen on the hematoxylin–eosin stain, and stage 1c with only portal/periportal fibrosis (Figure 2B). Stages 2–4 remained unchanged, including both perisinusoidal fibrosis and periportal fibrosis (Figure 2C) for stage 2, bridging fibrosis for stage 3 (Figure 2D), and cirrhosis for stage 4. The system is often applied to all cases of steatosis or steatohepatitis, irrespective of alcoholic or nonalcoholic, because the etiology is often unknown at the time of case sign-out.

## 3. Pathogenesis of NASH

Lipotoxicity in hepatocytes has been recognized as the fundamental feature of NASH [1]. The pathogenesis and mechanisms from simple steatosis to NASH have been broadly studied. In patients with NAFLD, lipid accumulation (particularly large droplet fat) (Figure 1A) can promote lipotoxicity and mitochondrial dysfunction in the hepatocytes, thus triggering hepatocyte apoptosis (Figure 1C), inflammation (Figure 1B) and pericellular to bridging fibrosis (Figure 2). Different lipids, free fatty acids (FFAs), and free cholesterol have been recognized as toxic species [25]. Indeed, recent findings regarding NASH pathogenesis and progression include: (1) NASH could be led by multiple hits through multiple intercellular interactions; (2) hepatocyte lipotoxicity and mitochondrial dysfunction (Figure 1D) induced by FFAs and their derivatives are the main drivers of liver injury; (3) decreased endoplasmic reticulum (ER) efficiency and increased ER stress could prolong unfolded protein response and hepatocellular apoptosis (Figure 1C); (4) upregulated proteins involved in multiple pathways and caspases could further result in mitochondrial dysfunction and apoptosis [26]. These have expanded our understanding of how NAFLD progresses to NASH, cirrhosis, and hepatocellular carcinoma (Figure 3). These important findings and our advancing understanding of mechanisms underlying NASH pathogenesis will be crucial for developing potentially important diagnostic biomarkers and effective target therapies. However, although the processes of hepatocyte lipotoxicity due to fat accumulation are quite clear, the mechanisms associated with the progression from simple steatosis to NASH and HCC are not fully characterized.

## 4. Role of Extracellular Vesicles and Exosomes in NASH Pathogenesis

Extracellular vesicles (EVs) are a heterogeneous group of small membrane vesicles secreted by cells to the extracellular environment in a highly regulated manner in normal or diseased conditions. The three major types of EVs include exosomes (30–120 nm in size), microvesicles (50–1000 nm in size), and apoptotic bodies (usually >500 nm in size) [17]. The liver is a large metabolism organ that consists of parenchymal cells, hepatocytes, and non-parenchymal cells. Effective communication and interaction between these different cells are crucial for proper physiological function and homeostasis. As types of EVs, exosomes and microvesicles are recently found to be important for cell-to-cell communication because they carry lipids, proteins, and various species of RNA that can trigger a myriad of responses in target cells. It is now well-established that EVs are ubiquitously released in the liver and participate in this process of NASH, particularly in promoting hepatocellular apoptosis, inflammation, and fibrogenesis [27,28,29].

### 4.1. Hepatocyte-Derived EVs in NASH

Hepatocytes are the major cell types of the liver, and 80% of the liver volume is contributed by the hepatocytes. The hepatocyte-derived EVs may play a critical role in the pathogenesis of NAFLD and NASH. In the context of NAFLD, excessive fatty acid accumulation is toxic to the hepatocytes, and this lipotoxicity can induce the release of EVs (hepatocyte-EVs) (Figure 4). The EVs carry bioactive components in their cargos and can modulate the metabolic changes in liver cells, which can mediate the progression of liver fibrosis [17]. Phagocytosis of the hepatocellular apoptotic bodies by Kupffer cells may potently increase the expression of TNF-α, TRAIL, and Fas ligand (FasL) [30] (Figure 4). These death receptor ligands could further induce apoptosis of the neighbor hepatocytes [31] (Figure 1C) and precipitate liver lobular inflammation (Figure 1B) and pericellular to bridging fibrosis (Figure 2). Lipotoxic hepatocyte-derived extracellular vesicles may add an additional link between hepatocellular injury and inflammation in NASH (Figure 1B). Mitochondrial DNA (mDNA) has been found to increase hepatocyte-derived EVs from mice and humans with NASH, which is implicated with characteristic sterile inflammation [32]. Lipotoxic endoplasmic reticulum (ER) stress activates IRE1A, which leads to the release of EVs from hepatocytes. These EVs are not only found to be enriched in ceramide but also sufficient to attract monocyte/macrophages into the liver [33,34]. Ibrahim et al. reported that lipotoxic hepatocyte-derived EVs also expressed C-X-C motif ligand 10 (CXCL10), a powerful Kupffer cell chemoattractant [35] (Figure 4). Bretz et al. reported that exosomes from body fluids such as liver cirrhosis ascites can induce a proinflammatory phenotype in monocytes [36]. Thus, the extracellular vesicles released from hepatocytes contain functional mediators of inflammatory response leading to lobular inflammation. Studies showed that palmitate-treated hepatocytes release proangiogenic microvesicles [37]. Importantly, the release of these vesicles with internalization into endothelial cells via Vanin-1 occurs before the onset of apoptosis and is associated with angiogenesis. Hepatocyte-derived EVs also have a well-documented role in promoting liver fibrosis via direct modulation of hepatic stellate cells (HSCs) phenotype. Povero et al. observed that lipid-induced hepatocyte-EVs interact with HSCs and subsequent upregulate profibrogenic genes, including collagen-1α1(Col1α1), α-smooth muscle actin (α-SMA), and tissue inhibitor of metalloproteinase-2 (TIMP-2) [38]. Hepatocyte-derived exosomal miR-27a and miR-1297 also promote profibrogenic activation in HSCs that contributes to liver fibrosis [39,40] (Figure 4).

### 4.2. Hepatic Non-Parenchymal Cell and Extra-Hepatic-Derived EV in NASH

Besides the hepatocytes, HSCs, sinusoidal endothelial cells (LSECs), Kupffer cells (KCs), and cholangiocytes are present in the liver and responsible for the regulation and support of hepatocyte activity. These non-parenchymal cells also secret EVs that have demonstrated contributions to the pathogenesis and progression of NASH. LSECs account for about 50% of non-parenchymal cells in the liver and are anatomically located near HSCs. Wang et al. found that LSECs-derived exosomes containing the sphingosine kinase 1 (SK1) protein can activate HSCs in vitro [41]. When HSCs are activated, they release exosomes containing CCN2 that promote hepatic fibrosis [42]. Cholangiocytes are also exosome-releasing cells. Studies have shown enrichment of the long non-coding RNA H19 in cholangiocyte-derived exosomes, which are delivered into hepatocytes, leading to hepatic lobular inflammation and biliary injury [43]. The crosstalk between hepatocytes and other cell types also plays an important role in the progression of NASH. Thus, EVs could be the prominent media. KCs produce endogenous miR-690 and, via exosome secretion, shuttle this miRNA to other liver cells, such as hepatocytes, recruited hepatic macrophages (RHMs), and HSCs. During the development of NASH, KCs become miR-690 deficient, and miR-690 levels are markedly lower in mouse and human NASH livers than in the controls. KC-specific knockout of miR-690 promotes NASH pathogenesis [44]. In addition, adipocyte-derived EVs have been demonstrated to promote hepatic inflammation via monocyte chemoattractant protein-1and IL-6 [45] and contribute to fibrosis via alteration of TIMPs and MMPs expression in hepatocytes and HSCs [46].

## 5. Circulating Biomarkers in Diagnosis of NASH

In clinical practice, the most important question regarding NAFLD/MAFLD is how to diagnose NASH early. Presently, although liver biopsy with histological evaluation is the golden standard for diagnosing and grading the disease of NASH, non-invasive options are desirable. In patients with NASH, most studies have confirmed that, despite several well-known limitations, transient elastography or point shear wave elastography can help in enriching the pool of patients that should be screened for investigational treatments [5,6]. For this reason, there are some new promising biomarkers that may help in diagnosing the early stages of NASH, but they are variables and not routinely tested. Regarding the possible non-biopsy predictors of NAFLD, several studies conducted by our group and others have evaluated a combination of serum/plasma biomarkers in NAFLD/NASH diagnosis [5,6]. PRO-C3 is a type III collagen marker reflecting levels of the N-terminal pro-peptide released by disintegrin and metalloproteases (ADAM)-TS2 during collagen maturation. PRO-C3 has been shown to be correlated with the fibrosis stage and additionally to be related to disease activity [5,47]. Thrombospondin (TSP) is a class of matricellular proteins that interacts with a number of ligands, including extracellular matrix structural proteins, cellular receptors, growth factors, and cytokines. TSP modulates cell–matrix interactions and possesses antiangiogenic properties [48]. However, evidence showed upregulation of the hepatic expression of the THBS2 gene, which encodes TSP2, has been reported in patients with advanced fibrosis compared with those without [49]. Clinical studies demonstrated that the serum TSP2 level correlated significantly with hepatocyte ballooning, lobular inflammation, and fibrosis stage in patients with NAFLD [50,51]. Simple non-invasive panels such as the NAFLD Fibrosis Score (NFS) and Fibrosis-4 (FIB-4) are recommended by the EASL-EASD-EASO Clinical Practice Guidelines as part of the diagnostic regimen for ruling out advanced fibrosis [52]. The guidelines further recommend the use of NFS and FIB-4 as prognostic markers to rule out progression to severe disease, including liver-related and all-cause mortality. Other multimarker models, such as the aspartate aminotransferase (AST)/platelet ratio index (APRI), are also used for fibrosis staging and prediction of liver-related events [53]. Reviewing the literature, we found other markers, such as the Enhanced Liver Fibrosis (ELF) test or FibroScan, had limited assessment for their prognostic ability. ADAPT, a PRO-C3-based fibrosis algorithm, performed better or equally well compared to APRI, FIB4, and AST/ALT ratio for detecting significant and advanced fibrosis but was superior in detecting and independently associated with NASH compared to APRI, FIB4, and AST/ALT ratio [54]. We recently conducted some clinical studies through the use of plasma biomarkers, including fragments of pro-peptides of type II, V, and VI procollagens for the detection of liver fibrosis in patients with type 2 diabetes [5,6]. We found that none of these non-biopsy tools assessed for the diagnosis of NASH in patients with type 2 diabetes had an optimum performance and none of the approaches did significantly better than the plasma level of aspartate aminotransferase (AST). We concluded that sequential use of plasma AST and other used non-biopsy tests may help in limiting the number of liver biopsies required to identify patients with advanced fibrosis [5]. To date, one biomarker alone is not able to identify patients with NASH and associated mild to moderate fibrosis. Since increased liver lobular inflammation is one of the major hallmarks of disease progression, liver inflammation-related circulating markers may represent an interesting source of early non-biopsy biomarkers for NAFLD and NASH. The proposed circulating markers could include cytokines, chemokines, or shed receptors from immune cells, circulating exosomes related to inflammation, and changing proportions of peripheral blood mononuclear cell (PBMC) subtypes [55].

## 6. Potential Roles of Exosomal miRNAs in Diagnosis and Treatment of NASH

NASH patients have a higher probability of developing cirrhosis and of dying from cardiovascular or liver-related causes [56,57]. Therefore, it is extremely important to identify NASH and fibrosis patients early among NAFLD patients in order to reduce mortality. Thus, several researchers attempted to develop non-invasive alternatives for the identification of patients at high risk of NASH and advanced fibrosis. Circulating miRNAs are stable, and techniques used for their detection are available [58]. MiRNAs circulating in the blood in a cell-free form have been recently reported to be related to the progression of NASH [23]. Analysis of these miRNAs may have a key role in biomarker discovery. So far, no current FDA-approved therapies have been used for the treatment of NASH. Recent studies have suggested that EVs may have the potential to develop therapeutic tools for treating NAFLD with their ability to stable delivery of drugs, miRNA, siRNA, or other cargoes and easy uptake by the liver cells [59]. Traditional therapies such as vitamin E and pioglitazone are promising for the improvement of steatosis and lobular inflammation; however, they have a limited effect on liver fibrosis. Diagnostic use of miRNAs is closer to clinical application than therapeutic use.

### 6.1. Potential microRNAs in Diagnosis of NASH

Animal models are frequently used to test the mechanisms of NAFLD and to find biomarkers for disease progression. Through experimental studies, various miRNAs are found to be important components of EVs leading to liver inflammation and fibrosis [17] (Figure 1B and Figure 2). Lee et al. reported that the hepatocytes treated with lipotoxic palmitate caused elevated levels of EVs and microRNAs (miRNA-122 and miRNA-192) (Table 1) that were implicated in driving steatohepatitis to liver fibrosis by upregulating the expression of fibrosis-related genes such as α smooth muscle actin (αSMA), collagen type 1 alpha 1 (col1α1), and transforming growth factor β (TGFβ) in HSC [60]. Consistent with this study, elevated levels of miR-122 and miR-192 were also observed in another study with mice fed with choline-deficient l-amino acid (CDAA), which is a diet that causes steatohepatitis [61]. In a clinical study, Becker et al. showed that elevated serum miRNA-122, -192, and -21 levels could be significantly correlated with NASH and performed as a biomarker in NASH patients [62]. Except for circulating miR-122 and miR-192 (Table 1), Tryndyak et al. found that mice liver histologic changes were associated with different levels of other hepatic and plasma miRNAs, including circulating miR-34a, miR-181a, and miR-200b (Table 1). The tested miRNA levels were found to be significantly correlated with the severity of NAFLD-specific liver histologic changes, while miR-34a showed the strongest correlation [61]. Broermann et al. recently studied the phosphodiesterase 5 (PDE5) inhibitor-induced effects on hepatic and plasma exosomal miRNA expression in CCl4-treated rats [63]. They found that in the livers of CCl4-treated rats, the expression of 22 miRNAs was significantly increased (>1.5-fold, *p* < 0.05), whereas the expression of 16 miRNAs was significantly decreased (>1.5-fold, *p* < 0.05). Importantly, the majority of the deregulated miRNA species were associated with liver fibrosis and inflammation, and the PDE5 inhibitor could suppress the induction of pro-fibrotic miRNAs, such as miR-99b, miR-100, and miR-199a-5p, and restore antifibrotic miR-122 and miR-192 in the liver [63]. These findings suggested that miRNA profiling of plasma exosomes might be used as a biomarker for NASH progression and monitoring the treatment effects.

Clinically, Ezaz et al. recently studied the differential associations of circulating miRNAs, including miR-34a, miR-122, miR-191, miR-192, and miR-200a with histopathologic changes and some clinical factors in 132 liver biopsies confirmed NAFLD patients. They found that miR-34a, miR-122, miR-192, and miR-200a (Table 1), but not miR-191, strongly correlated with the stages of fibrosis. From their multivariate analysis, miR-34a, miR-122, and miR-192 levels were independently associated with liver steatosis and fibrosis but not lobular inflammation or ballooned hepatocytes, whereas miR-200a was only associated with liver fibrosis. Among them, miR-34a has the strongest predictive value for liver fibrosis stages [64] (Table 1). Murakumi et al. performed a comprehensive miRNA level evaluation in the peripheral blood of 64 patients with liver diseases, including NASH, and found that miRNA expression pattern in exosome-rich fractionated serum showed high potential as a biomarker for diagnosing liver diseases including NASH [65]. Another recent study conducted by Muhammad Yusuf et al. showed the expression levels of selected miRNAs (miR-182, miR-301a, and miR-373) in exosomes of the serum and ascitic fluid in 52 patients with NASH-caused cirrhosis with or without associated HCC using quantitative real-time PCR (Table 1). Their preliminary data showed a significant increase in the expression levels of exosomal miR-182, miR-301a, and miR-373 in both serum and ascetic fluid, suggesting the possible roles of these miRNAs as circulating biomarkers for NASH-caused cirrhosis and associated HCC [66] (Table 1). Interestingly, Jampoka et al. found that decreased serum level of miR-29a (Table 1) in NASH patients are strongly negatively correlated with the disease of NASH [67]. These studies show the important roles of exosomal miRNAs in the pathogenesis of NASH that shed light on developing potential important biomarkers for future clinical diagnosis of NASH through more well-designed clinical studies.

### 6.2. Potential Exosomal miRNA in Future Treatment of NASH

Liver cirrhosis caused by NASH can lead to HCC (Figure 3) and has high morbidity and mortality, with only liver transplantation as the therapeutic option. Importantly, recent studies have suggested that EVs may have the potential to develop therapeutic tools for treating NAFLD with their ability to stable delivery of drugs, miRNA, siRNA, or other cargoes and easy uptake by the liver cells [17]. Indeed, more recent experimental studies have demonstrated EVs as potential therapeutic targets and inhibition of EV-mediated pathological processes can block disease progression in NAFLD [68,69]. Since exosomes, the smaller size of EVs, may have important roles in the crosstalk between hepatocytes and hepatic stellate cells in the progression of NASH [60], treatment targeting the exosomal miRNAs may help develop targeted therapy for NASH-caused cirrhosis and/or HCC.

It’s known that exosomal microRNA-223 (miR-223) has an antifibrotic effect [70], and NAFLD is associated with some elevated cytokines, particularly IL-6. Hou et al. examined the serum IL-6, specific IL-6 receptor A (sIL-6Ra), and miR-223 levels in patients with NAFLD/NASH and found that elevated serum IL-6 and miR-223 levels were correlated with each other in patients with NAFLD. Through an in vivo study with a knockout (KO) mice model, they showed evidence of how hepatocyte and myeloid-specific IL-6 signaling affects the pathogenesis of NAFLD by regulating the antifibrotic miR-223. Interestingly, they found that IL-6 KO mice developed worse liver fibrosis, and the increased liver fibrosis in myeloid-specific Il6ra KO mice was most likely attributable to the reduction of antifibrotic miR-223 level leading to up-regulation of the miR-223 target gene TAZ (transcriptional activator with PDZ-binding motif) that is a well-known factor to promote NASH fibrosis. In their in vitro experiments, they found that IL-6 treatment upregulated exosome biogenesis-related genes leading to more release of miR-223-enriched exosomes by macrophages [70]. These results could suggest that inhibition of myeloid-specific IL-6 signaling may lead to a decrease in liver fibrosis through exosomal transfer of antifibrotic miR-223 into hepatocytes, providing potentially important therapeutic targets for NASH treatment. In the latest article, Gao et al. documented that KCs-derived miR-690 have crosstalk with other liver cells, such as hepatocytes, RHMs, and HSCs. MiR-690 directly inhibits fibrogenesis in HSCs, inflammation in RHMs, and de novo lipogenesis in hepatocytes. When a miR-690 mimic is administered to NASH mice in vivo, all the features of the NASH phenotype are robustly inhibited. These studies show that KCs may play a central role in the etiology of NASH and raise the possibility that miR-690 could emerge as a therapeutic potential for this condition [44].

Another example is about endoplasmic reticulum to nucleus signaling 1 (ERN1, also called IRE1A) that is activated in the liver tissue of patients with NASH. Activation of IRE1A hepatocytes can release ceramide-enriched inflammatory EVs [33]. Dasgupta et al. recently demonstrated the effects of inhibiting IRE1A by intraperitoneal injections of the IRE1A inhibitor 4μ8C on the release of inflammatory EVs in the mice with NASH leading to hepatocyte-specific disruption of IRE1A. They also found that activated IRE1A promotes the transcription of serine palmitoyltransferase genes via XBP1, resulting in ceramide biosynthesis and the release of EVs in the hepatocytes from mice. Interestingly, the levels of XBP1, serine palmitoyltransferase, and EVs are all increased in the liver tissues of patients with NASH [33]. These findings raise some strategies to target those EVs that may reduce liver inflammation and have a benefit on patients with NASH. However, there is no direct evidence showing those EVs contain miRNAs. More investigations are needed.

Since exosomal miRNAs have the ability to increase collagen synthesis and angiogenesis, they may play a major role in proper tissue remodeling and extracellular matrix degradation in patients with NASH fibrosis. Various studies have demonstrated the effect of exosomes on improving the outcome of cutaneous wound healing, scar tissue formation, and degenerative bone disease [71]. Through an experimental NASH animal model, El-Darany et al. recently studied the therapeutic potentials of bone marrow mesenchymal stem cells (BM-MSCs) and their derived exosomes (BM-MSCs-Exo). They found that significant upregulation in fatty acid oxidation (PPARα, CPT1) was associated with the abrogation of steatosis and ballooned hepatocytes in HFD-caused NASH. Treatment with BM-MSCs or co-treatment with BM-MSCs-Exo revealed significant anti-apoptotic effects mediated by a significant decrease in Bax/Bcl2 ratio. In addition, a significant increase in mitochondrial mitophagy gene expressions, including Parkin, PINK1, ULK1, BNIP3L, ATG5, ATG7, and ATG12, were detected in BM-MSCs treated or BM-MSCs-Exo co-treated groups. These changes are thought to be modulated by the upregulation of miRNA-96-5p, leading to the downregulation of its downstream target caspase-2 [71]. Being a critical player in NASH progression, targeting caspase-2 by miRNA-96-5p could be a promising therapeutic modality to treat NASH. More studies, particularly clinical studies, are needed.

**Table 1 ijms-23-13501-t001:** Deregulated circulating miRNAs in the blood of the patients with NASH.

miRNA	Disease-Associated Change	Reference
miR-122	↑	[62,67,72,73,74]
miR-192	↑	[62,72,73]
miR-21	↑	[62,75]
miR-34a	↑	[73,74,75]
miR-181a	↑	[76,77]
miR-200a	↑	[64]
miR-182	↑	
miR-301a	↑	
miR-373	↑	
miR-29a	↓	[67]

↑ and ↓ arrows designate upregulation and downregulation, respectively.

## 7. Future Directions

NASH is the histologic manifestation of the wound-healing response to hepatocyte lipotoxicity due to steatosis (particularly large droplet fat accumulation), lobular inflammation, and ballooning degeneration of the hepatocytes. The outcomes of steatosis and associated lipotoxicity could be determined by the type and amount of lipid that accumulates, as well as the ability of hepatocytes to defend against or adapt to the accumulation the lipids. These might be prevented or treated by preventing or reduction of lipotoxicity. The repair process of NAFLD/NASH is complex, and deregulated healing responses will worsen clinical outcomes by promoting the development of cirrhosis and hepatocellular carcinoma. Through more experimental studies in NASH pathogenesis and clinical studies, exosomal miRNA expression profiles in exosome-rich fractionated serum could be useful for determining the disease progression and developing potentially important biomarkers for future clinical diagnosis, predicting prognosis, and targeted therapy in patients with NASH.

## 8. Conclusions

NAFLD has become the most common chronic liver disease in the US. Patients with progression from NASH and advanced fibrosis were associated with increased risks of HCC. In recent years, EVs in the field of metabolic liver disease have gathered significant traction. During lipotoxicity, EVs are released in large quantities from different hepatocyte or hepatic non-parenchymal cells. These EVs play a key role in the transmission of information between different cell types in the liver. They can change the function of target cells and activate different pathways, promoting the pathogenesis and development of NAFLD/NASH. Through mouse model experimental studies and limited clinical observations, various exosomal miRNAs are associated with liver steatosis, hepatocyte ballooning, inflammation, and fibrosis. A panel of exosomal and other EV miRNAs listed in Table 1 could be developed as potential circulating biomarkers for future clinical diagnosis of NASH through more well-designed clinical studies. Clinical studies targeting the exosomal miRNAs, such as miR-96-5p, may help develop targeted therapy for NASH cirrhosis or HCC. With more well-designed clinical studies and RCTs, exosomal miRNAs may be demonstrated to be useful in the clinical diagnosis and treatment of NAFLD/NASH in the near future.

## Figures and Tables

**Figure 1 ijms-23-13501-f001:**
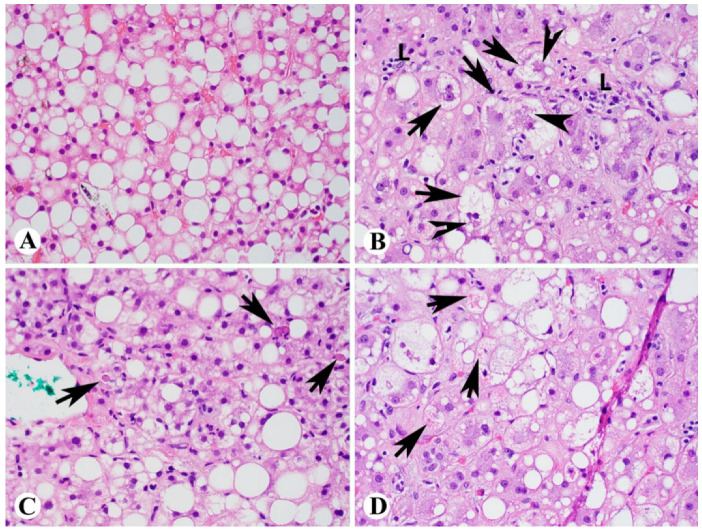
Histology of steatosis and steatohepatitis. (**A**): Macrovesicular steatosis; (**B**–**D**): steatohepatitis showing ballooned hepatocytes (arrows) with Mallory–Denk bodies (arrowheads), lobular inflammation (L), and mild perisinusoidal/pericellular fibrosis (**B**); acidophilic bodies/hepatocytes apoptosis (arrows) (**C**); and megamitochondria (arrows) (**D**) ((**A**–**D**), H&E stain, 400×).

**Figure 2 ijms-23-13501-f002:**
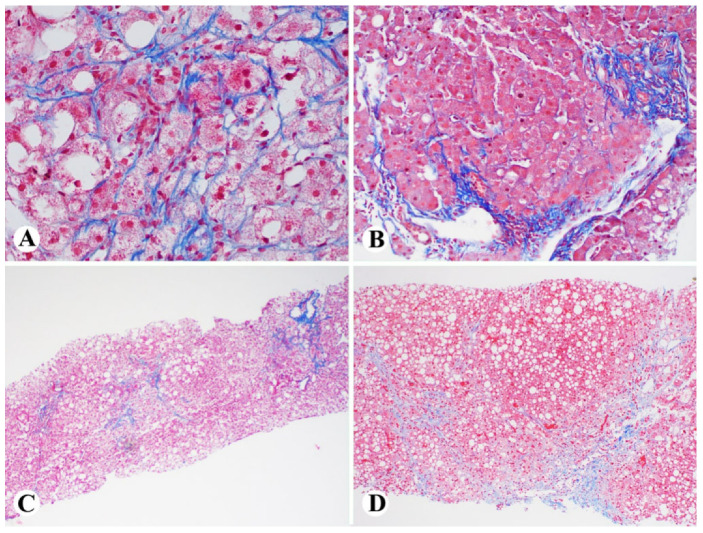
Fibrosis staging of steatohepatitis: (**A**–**D**), Trichrome stain highlighting the perisinusoidal/pericellular fibrosis (**A**); periportal fibrosis (**B**); perisinusoidal/pericellular fibrosis and periportal fibrosis (Stage 2, (**C**)) and bridging fibrosis (stage 3, (**D**)) ((**A**), 400×; (**B**), 200×; (**C**,**D**), 100×).

**Figure 3 ijms-23-13501-f003:**
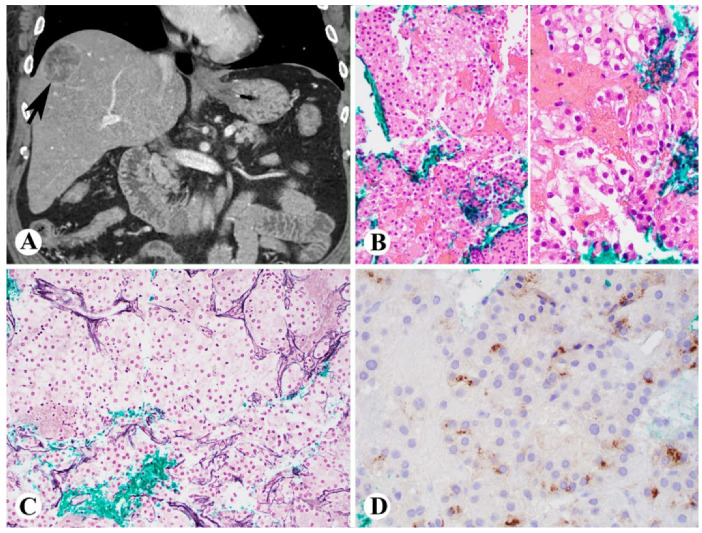
Steatohepatitic type of hepatocellular carcinoma (HCC) in a 60-year-old female with history of nonalcoholic steatohepatitis. (**A**), Computer tomography scans showing a 4.8 cm low-attenuation mass (arrow, coronal); (**B**–**D**), Liver needle core biopsy showing well-differentiated HCC (**B**); loss of reticulin framework (**C**); and focal positivity of glypican 3 ((**B**), H&E stain, left 100×, and right 400×; (**C**), reticulin stain, 100×; (**D**), immunohistochemistry for glypican 3, 400×).

**Figure 4 ijms-23-13501-f004:**
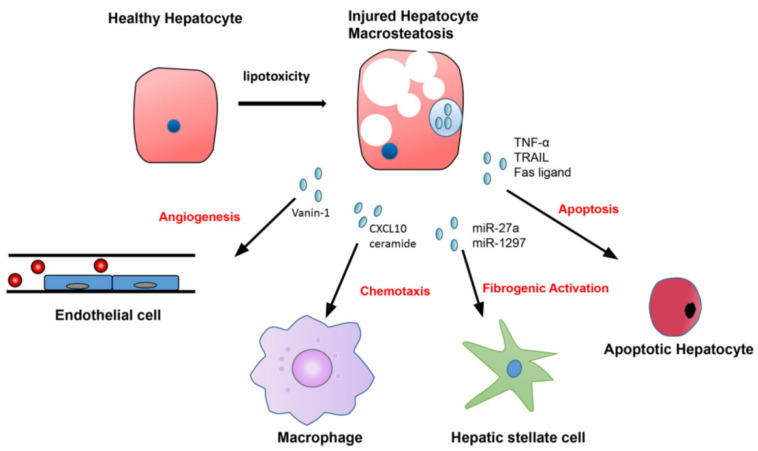
Lipotoxic hepatocytes release numbers of EVs containing specific cargo that contribute to disease processes in NASH. Accumulation of excessive fatty acid in hepatocytes promotes the release of EVs with cargo that promote Vanin-1 mediated endothelial cell angiogenesis; CXCL10 and ceramides-mediated macrophage chemotaxis; microRNA (such as miR-27a and miR-1297) mediated hepatic stellate cell activation, and TNF-a TRAIL and Fas ligand mediated apoptosis of the neighbor hepatocytes.

## Data Availability

Data supporting the reported results can be provided by the corresponding author.

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
