# Peer review of "Exosomal microRNAs and Progression of Nonalcoholic Steatohepatitis (NASH)"

_ijms, 2022, doi:10.3390/ijms232113501_

Round 1
Reviewer 1 Report
In this manuscript, Qi et al. described the roles of miRNA in NASH pathogenesis and progression, as well as their potential in future clinical diagnosis and treatment for patients with NASH. However, this reviewer has the following concerns.
Major comments:
1. This manuscript does not explain the relationship between extracellular vesicles (EV) and microRNAs (miRNAs). It is important whether only miRNAs or EV which includes miRNAs and other components are crucial for developing NASH because several factors in EV are described in Figure 4.
2. Regarding miRNAs raised in Table 1, none of them appear in Figure 4. Although miRNAs in Table 1 are circulating, they should have a function in target cells.
3. Current biomarkers for NASH should be more precisely mentioned. At least, PRO-C3, Thrombospondin-2, ELF, FIB-4, NIS4, ADAPT, etc. should be described.
4. In 4.2. Hepatic Non-Parenchymal Cell and Extra-Hepatic-Derived EV in NASH, sinusoidal endothelial cells (LSECs), Kupffer cells, and cholangiocytes are described in the text, but only endothelial cells (not specified as LSEC) appear in Figure 4.
5. In 6.2. Potential Exosomal miRNA in the future treatment of NASH, disruption of IRE1A by 4m8c is described, but miRNA has nothing to do with this action.
6. In 6.1. Potential microRNAs in the diagnosis of NASH and 6.2. Potential Exosomal miRNA in the future treatment of NASH, it should be described that diagnostic use of miRNAs is closer to the clinical application than therapeutic use and how clinical trials are going on as far as possible.
Minor comments:
On page 1, line 31, “potentialities” should be “potential”
Reviewer 2 Report
This is a review covering the role of exosomes in NAFLD/NASH pathogenesis and their potential use in the diagnosis and treatment of the disease. The review fits one of the scopes of the journal (“fundamental theoretical problems of broad interest in biology, chemistry and medicine”) and treats a really interesting topic that may yield better understanding of mechanisms involved in NAFLF/NASH pathophysiology. However, there are some confusing information and the conclusion section needs to be clarified and improved. Here, we provide some advice, and a number of major and minor points for revision are listed below.
Title and abstract
- The title is appropriate for the content of the manuscript.
- The abstract should be improved. It is confusing for the readers. The objective of the manuscript is not clear. Do the authors review the role of exosomes in NAFLD/NASH or the updated knowledge about the pathophysiology of the disease?
Major points
1. In line 51, the authors state “So far, no current FDA-approved therapies are used for the treatment of this disease and to find appropriate therapeutic targets is now more urgent than ever before.”. This statement should be supported by a more recent reference. See S. Rana et al. Current treatment paradigms and emerging therapies for NAFLD/NASH, 2021, doi: 10.2741/4892.
2. In lines 53-56, the authors talk about the results of traditional therapies such as vitamin E. However, the reference (12) is not up-to-date and it does not give enough information. I suggest checking papers that provide a more exact view of the effect of these drugs in the improvement of fibrosis in NASH, for example, Majzoub et al.: Systematic review with network meta-analysis: comparative efficacy of pharmacologic therapies for fibrosis improvement and resolution of NASH, 2021, doi: 10.1111/apt.16583. In this study, the authors conclude that “compared to placebo, Lanifibranor, Obeticholic acid, Pioglitazone and Vitamin E were statistically significantly better in achieving ≥1 stage of fibrosis improvement.".
3. In line 66, authors state “miRNAs are short, non-coding RNA species that regulate lipid metabolism, inflammation, and fibrosis in NAFLD”. Other functions of miRNAs should be commented. The reader needs first a more general description of miRNAs.
4. In line 157, authors state that “Recent studies on NASH in mouse models and human subjects have demonstrated that hepatocytes are the major contributors to the increased levels of circulating EVs”. Are hepatocytes the major contributors of EVs because they are the majority cell type in the liver, or the levels of hepatocyte-derived EVs are independent on the number of cells?
5. In lines 194-199 the authors explain exosomes’ release by different hepatic cells and the interaction between the exosomes and hepatocytes. This information should be included in Figure 4.
6. Table 1 should include information about miR-181a and miR-200b.
7. In table 1, it would be appropriate to describe the information the arrows are supposed to provide.
8. Conclusion section of the review should be improved since it virtually contains the same information of the abstract.
Minor points
1. The text has some typos (e.g., lines 66, 69, 293…). Please correct.
2. Some abbreviations and their meanings are not indicated in the text. Example: “CRN” appears for the first time in line 48, but the acronym’s meaning is explained in line 84.
Round 2
Reviewer 1 Report
All the comments have been addressed.